# Ultra-High-Sensitivity, Miniaturized Fabry-Perot Interferometric Fiber-Optic Microphone for Weak Acoustic Signals Detection

**DOI:** 10.3390/s22186948

**Published:** 2022-09-14

**Authors:** Guojie Wu, Haie Li, Hongxin Ye, Zhenfeng Gong, Junsheng Ma, Min Guo, Ke Chen, Wei Peng, Qingxu Yu, Liang Mei

**Affiliations:** 1School of Optoelectronic Engineering and Instrumentation Science, Dalian University of Technology, Dalian 116024, China; 2School of Mechanical Engineering, Dalian University of Technology, Dalian 116024, China; 3School of Physics, Dalian University of Technology, Dalian 116024, China

**Keywords:** weak acoustic signals detection, Fabry-Perot, fiber-optic microphone (FOM)

## Abstract

An ultra-high-sensitivity, miniaturized Fabry-Perot interferometric (FPI) fiber-optic microphone (FOM) has been developed, utilizing a silicon cantilever as an acoustic transducer. The volumes of the cavity and the FOM are determined to be 60 microliters and 102 cubic millimeters, respectively. The FOM has acoustic pressure sensitivities of 1506 nm/Pa at 2500 Hz and 26,773 nm/Pa at 3233 Hz. The minimum detectable pressure (MDP) and signal-to-noise ratio (SNR) of the designed FOM are 0.93 μPa/Hz^1/2^ and 70.14 dB, respectively, at an acoustic pressure of 0.003 Pa. The designed FOM has the characteristics of ultra-high sensitivity, low MDP, and small size, which makes it suitable for the detection of weak acoustic signals, especially in the field of miniaturized all-optical photoacoustic spectroscopy.

## 1. Introduction

Weak acoustic signal detection is essential in petroleum exploration [1], gas leakage monitoring [2], and photoacoustic spectroscopy (PAS) [3]. Especially in miniaturized all-optical PAS [4,5,6] field, the microphones, featuring high sensitivity, a high signal-to-noise ratio (SNR), ultimately improve the detection limit of trace gases. Fiber-optic microphones (FOMs) featuring high sensitivity, high SNR, high resolution, and immunity to electromagnetic interference, have become one of the most efficient solutions for weak acoustic signal detection [7,8,9,10]. The common FOM structures include a Mach-Zehnder interferometer (MZI) [11], a Michelson interferometer (MI) [12,13,14], and a Fabry-Perot (F-P) interferometer [15,16,17]. Among the above FOMs, the F-P interferometric (FPI)-based FOMs have attracted extensive attention from researchers due to their lower cost, smaller volume, higher sensitivity, and greater stability. The diaphragm is forced to vibrate via external sinusoidal acoustic pressure. Thus, the performance of the diaphragm, depending on its material and construction [18], is crucial for the practical application of the FOMs. Recently, various diaphragm materials have been employed on FPI-based FOMs to improve their performance in practical applications, e.g., silver, silicon, Parylene-C, polymer, and graphene [19,20,21,22,23,24,25,26]. In 2015, Li et al. [27] fabricated a FOM using multilayer graphene films (diameter = 125 µm) and achieved a sound pressure sensitivity of 2380 nm/kPa. In 2018, Ni et al. [25] reported a FOM based on an ultra-thin graphene diaphragm (thickness = 10 nm). A minimum detectable pressure (MDP) of 33.97 μPa/Hz^1/2^ at 10 kHz was achieved. Although the performance of graphene-based FOMs was better, there were some disadvantages, such as a complex manufacturing process and high costs. Xu et al. [7] demonstrated a FOM utilizing a large area of silver film. The sensitivity and MDP reached 160 nm/Pa and 14.5 μPa/Hz^1/2^ at 4 kHz, respectively. In 2017, we proposed a highly sensitive FOM based on a thin silver diaphragm [9]; the pressure sensitivity, temperature sensitivity, and MDP were 5.97 µm/Pa, 0.139 nm/°C, and 40 μPa/Hz^1/2^ at 1.6 kHz. Despite the fact that the manufacturing process of silver-film-based FOM was simple, a large-area diaphragm was required to improve the sensitivity, which limited its application range.

Recently, several novel diaphragm structures have attracted many researchers, who applied them to FPI-based FOMs. In 2020, Yao et al. [28] developed an ultra-small fiber-tip FOM utilizing optical three-dimensional printing technology. The sensitivity and MDP reached 118.3 mV/Pa and 0.328 μPa/Hz^1/2^ under experimental tests. Although the fiber-tip FOM was small in size and had excellent sensitivity and MDP, it had some drawbacks, such as complex manufacturing processes and a poor SNR that limited its practical application in industry. Chen et al. [29] proposed a highly sensitive and stable FPI-based FOM by combining a white-light interference (WLI) demodulation algorithm and a cantilever based on a stainless steel material. This FOM, with sensitivity and MDP corresponding to 211.2 nm/Pa and 5 μPa/Hz^1/2^ at a working frequency of 1 kHz, had been applied to an all-optical PAS for trace gas detection. However, due to the precision of the laser-marking machine and the characteristics of stainless steel, the sensitivity of the cantilever-based FOM was very low when its operating frequency exceeded 2000 Hz. Therefore, it was not suitable for use in high-frequency scenarios. In 2021, we presented an FOM based on a small-size cantilever with a sensitivity of 950 nm/Pa at 14.8 kHz, an MDP of 25.68 μPa/Hz^1/2^ at 13 kHz, and an SNR of 71.81 at 0.1 Pa [30]. Although the FOM was small in size, it had poor sensitivity and SNR. Moreover, in order to adapt the working frequency of the silicon cantilever, an ultra-high-speed demodulator was required, which increased the demodulation system cost.

In this paper, we report an ultra-high-sensitivity, miniaturized FPI FOM for weak acoustic signal detection. The FOM includes a silicon cantilever, a copper material shell, and a single-mode fiber. The working frequency of the silicon cantilever is less than a few kHz, avoiding the use of an ultra-high-speed spectrometer. A miniaturized FOM as the acoustic wave detection unit and a high-speed spectrometer as the demodulator have been successfully implemented for weak acoustic signal detection.

## 2. Sensor Design and Simulation

The FOM design, based on an FPI structure, is presented in Figure 1, and mainly includes a silicon cantilever, a copper material shell, and a single-mode fiber. The two reflecting end-faces of the F-P interference are composed of the fiber end-face and the cantilever, respectively. The cantilever has a dimension of 1.32 mm × 0.8 mm × 3.9 μm (length × width × thickness). The copper material shell is a cylinder with a diameter of 3.6 mm and a length of 10 mm. The volume of the inner cavity is 60 cubic millimeters. Since the sensitivity of the cantilever transducer directly affects the performance of the FOM, it becomes particularly critical to analyze the factors affecting the cantilever. When the applied sinusoidal pressure wave acts on the proposed cantilever-based FOM, the theoretical frequency response of the cantilever is usually described as reported in [31]:(1)$$R_{c}(f)=\frac{\omega \tau_{2}}{m\sqrt{{\left(\omega_{0}{}^{2}-\omega^{2}\right)}^{2}+{\left(\omega /m\right)}^{2}}\sqrt{1+{\left(\omega \tau_{2}\right)}^{2}}}$$
where m, ω, and $\omega_{0}$ are the effective mass of the cantilever, angular frequency, and first-order resonant angular frequency. $\tau_{2}$ can be found in [31]. $f_{0}$ can be expressed as:(2)$$f_{0}=\frac{1}{2\pi }\sqrt{\frac{k}{m}}=\frac{1}{2\pi }\sqrt{\frac{\frac{2}{3}Ew{\left(\frac{d}{l}\right)}^{3}+\frac{\gamma \;pA_{c}{}^{2}}{2.5V}}{0.647m_{c}+V\rho }}$$
where k and E are the effective elastic modulus and Young’s modulus of the cantilever, respectively. $m_{c}$, d, l, ω, and ρ are the effective mass, thickness, length, width and density of the cantilever, respectively. V represents the volume of the air cavity.

According to Equations (1) and (2), when the size of the cantilever is determined, the volume of the cavity will also affect the resonant frequency and sensitivity of the FOM.

Figure 2 presents the finite element analysis of the proposed FOM vibration modes by utilizing the COMSOL software. The simulation analysis uses mainly solid mechanics, thermo-viscous acoustics, and pressure acoustics modules. An external acoustic pressure field of 1 Pa is used to generate a cantilever response corresponding to the acoustic wave. Figure 2 shows the vibration pattern of the cantilever at the first-order resonance frequency (external sound pressure = 1 Pa), which shows that the end-face of the cantilever has the maximum deflection. Meanwhile, the simulation frequency response of the designed FOM is shown in Figure 3, indicating that the maximum sensitivity of the cantilever is 29,036 nm/Pa, corresponding to the first-order resonance frequency (3340 Hz).

The two reflecting end-faces of the F-P interference are composed of the fiber end-face and the cantilever, and the air gap between them constitutes the F-P cavity. When a sinusoidal sound pressure is applied to the designed FOM, the length of the F-P cavity will be periodically changed. Due to the fact that the reflectivity of the fiber end-face is only about 4%, the FPI-based FOM is generally described as a two-beam interferometer. In addition, the demodulation algorithm, based on WLI and featuring high demodulation accuracy, strong anti-interference ability, and fast demodulation speed, is one of the most effective methods for the demodulation of F-P cavity length. The interference spectrum of the F-P FOM, based on the WLI algorithm, is generally described as given in [32]:(3)$$I(\lambda )=2I_{0}(\lambda )[1+\gamma \cos(\frac{4\pi (\Delta d+d_{0})}{\lambda }+\pi )]$$
where $I_{0}$ is the intensity of the white light source, γ is the fringe fineness, $d_{0}$ and Δd represent the static cavity length and the cavity-length variation of the F-P cavity, respectively.

## 3. Experimental Sections and Results Analysis

### 3.1. Experimental Setup

The experimental setup for the frequency response of the proposed FOM is presented in Figure 4. A super-luminescent diode (SLD) with a parameter of 1550 nm (center wavelength) × 60 nm (bandwidth) serves as the broadband light source. An emitting light is injected into the circulator through a fiber and then propagates into the designed FOM. A sine wave from a loudspeaker driven by a lock-in amplifier forces the cantilever to deform, resulting in its periodically changing the F-P cavity length. The reflected interference light is fed back to the high-speed spectrometer (FBGA analyzer, BaySpec, San Jose, CA, USA). The acquired interference spectrum is processed by a computer using the high-speed WLI demodulation algorithm, and the corresponding cavity length variation is obtained. The electrical condenser microphone of the B&K 4189 (sensitivity = 45.7 mV/Pa) is placed near the designed FOM for acoustic calibration. The output voltage of the B&K 4189 is recorded by a data acquisition instrument (DAQ) (ENET-9163, National Instruments, Austin, TX, USA) and transmitted to a computer.

During the acoustic test, the two microphones (the designed FOM and the B&K 4189) and the loudspeaker are placed in a soundproof cabinet to reduce the effect of external ambient noise on the frequency response test results.

### 3.2. Experimental Results Analysis

Figure 5 depicts the interference spectrum of the designed FOM, collected by the high-speed spectrometer. It can be seen from Figure 5 that the fabricated FOM exhibits a good interference contrast, which ensures a good SNR in the subsequent demodulation process. The average static F-P cavity length is calculated to be 707.2 μm. Under the conditions where the sound pressure is adjusted to 0.003 Pa, the time domain responses of the designed FOM at frequencies of 1 kHz, 2 kHz, 2.5 kHz, and 4 kHz are presented in Figure 6. It demonstrates that the designed FOM operates smoothly at different acoustic frequencies.

By controlling the loud-speaker output sound frequencies in the range of 1000–4500 Hz, the actual frequency response of the designed FOM is recorded (see Figure 7), showing an upward trend first and then a downward trend, and the upward trend is more obvious after 3100 Hz and the peak value appears at the first-order resonant frequency (3233 Hz) of the FOM. The designed FOM sensitivity reaches 26,773 nm/Pa at the first-order resonant frequency. Meanwhile, the Q value reaches about 65 (3233/50). The frequency response curve shows a relatively flat trend from 1000 Hz to 2700 Hz. The designed FOM not only has high sensitivity but also has a high Q value, so the sensitivity in the relatively flat frequency response region (2500 Hz) also reaches 1506 nm/Pa, which is better than the sensitivity of the most currently reported FOM at the resonance frequency.

In order to verify the correctness of the conclusions obtained from Equations (1) and (2), two other FOMs with the same cantilever size but different cavity volumes are fabricated. By repeating the above experimental process, Figure 8 is obtained. The comparative data in Figure 7 and Figure 8 are presented in Table 1. It can be seen that with the increase in the cavity volume, the resonance frequency of the FOM decreases, and the sensitivity and the Q-value increase.

The long-term stability of the FOM is an important indicator to evaluate whether the sensor can be practically applied. Therefore, we have performed a test of this characteristic on the designed FOM. The sensitivity data at operating frequencies of 1 kHz, 2 kHz, 2.5 kHz, and 4 kHz is recorded by setting the measurement time for each data point to 1 s and performing continuous measurements (see Figure 9). The long-term sensitivity of the FOM has little in the way of fluctuation range at different operating frequencies. In order to analyze the long-term stability of the FOM at different frequencies more clearly, we perform statistical calculations on the measurement results (see Table 2). When the working frequency is 2.5 kHz, the average sound pressure sensitivity of the FOM reaches 1506.1 nm/Pa and the deviation at this time is ±1.9%. When the frequencies are 1 kHz, 2 kHz, and 4 kHz, the corresponding deviations are ±1.4%, ±1.1%, and ±2.3%, respectively. Through long-term stability analysis, it can be seen that the FOM can work stably at 1 kHz, 2 kHz, 2.5 kHz, and 4 kHz.

The linear response to different acoustic pressures is critical for testing FOM performance. Therefore, under the sound pressure range of 0.001–1 Pa from a loud-speaker with a frequency of 2.5 kHz, the output amplitudes of the designed FOM under different sound pressures are shown in Figure 10. It can also be seen from Figure 10c that the sensitivity has better stability under different sound pressures Through the linear fitting method, it can be known that the designed FOM has a high sensitivity (value = 1506.3 nm/Pa) and a good linear response (R^2^ = 0.9999) at 2500 Hz.

Figure 11 depicts the amplitude spectrum of the FOM when a sound pressure of 0.003 Pa is applied at 2.5 kHz. It shows that the floor noise and the SNR are −109.78 dB and 70.17 dB for a 1 Hz resolution bandwidth. The MDP of the designed FOM, representing the ability to detect ultraweak sound pressure, is 0.93 μPa/Hz^1/2^. Table 3 summarizes the performance of other previously developed FPI-based FOMs for comparison. Compared with the FOMs mentioned in Refs. [7,18,19,33], a more stable and anti-interference WLI demodulation algorithm is used that provides a guarantee for the stable operation of the system. In summary, compared to the previously developed FOM, the FOM designed in this paper achieves ultra-high sensitivity and ultra-low MDP while maintaining a small volume, which is suitable for the detection of weak sound signals.

## 4. Conclusions

In conclusion, an ultra-high sensitivity, miniaturized FOM has been developed that utilizes a silicon cantilever as an acoustic transducer. The cantilever has a dimension of 1.32 mm × 0.8 mm × 3.9 µm. The effect of cavity volume on FOM sensitivity is analyzed by theory and experimental analysis, showing that the effect of a larger cavity volume on FOM is a decrease in the resonance frequency and an increase in the sensitivity and Q value. The volumes of the cavity and the FOM are finally determined to be 60 and 102 cubic millimeters, respectively. The first resonance frequencies of the cantilever are 3233 Hz and 3340 Hz from the experiment and simulation, showing good consistency. The results of the experimental tests show that the FOM has acoustic pressure sensitivities of 1506 nm/Pa at 2500 Hz and 26,773 nm/Pa at 3233 Hz. The MDP and SNR of the designed FOM are 0.93 µPa/Hz^1/2^ and 70.14 dB at an acoustic pressure of 0.003 Pa. Meanwhile, the proposed FOM has an approximately linear value (R^2^ = 0.9999) under the applied sound pressure range of 0.001–1 Pa. Compared with the previously developed FOM, the FOM designed in this paper achieves ultra-high sensitivity and ultra-low MDP, while maintaining a small volume. The FOM developed in this work shows great potential for the detection of weak sound signals in a small space.

## Figures and Tables

**Figure 1 sensors-22-06948-f001:**
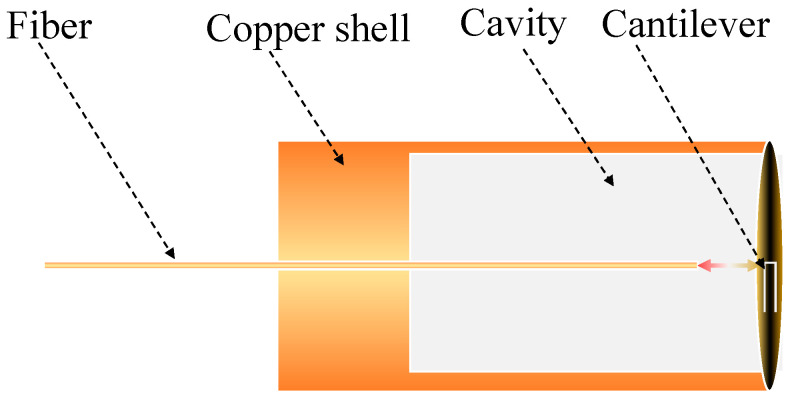
The schematic diagram of the FOM design, based on the FPI structure.

**Figure 2 sensors-22-06948-f002:**
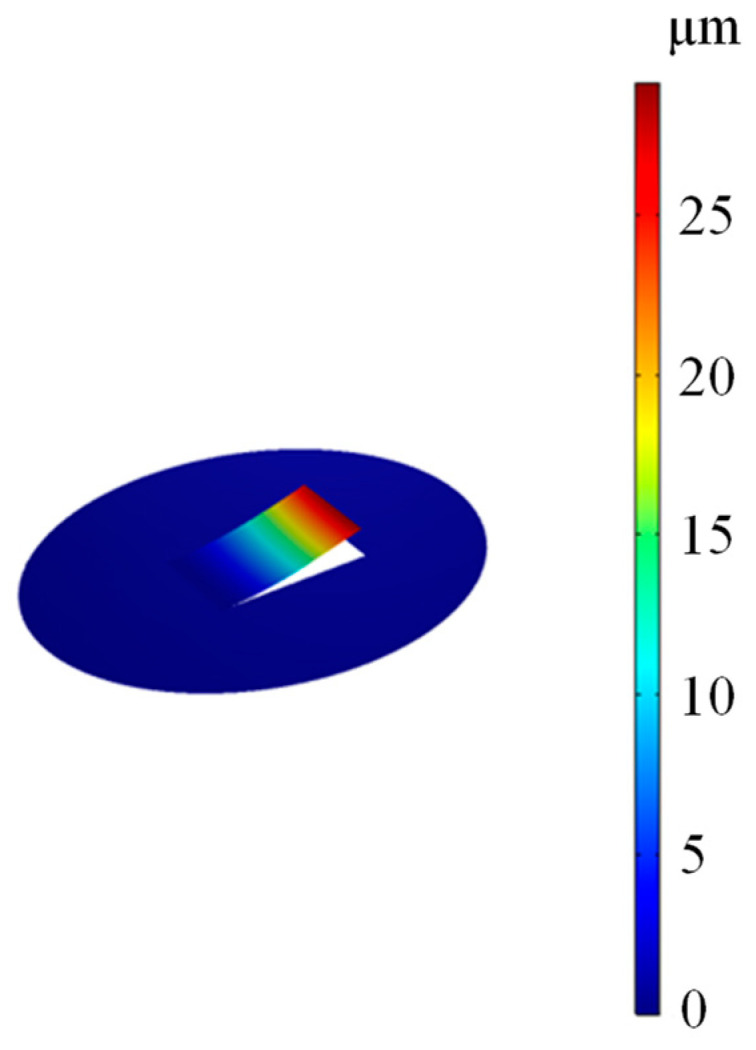
The vibration pattern of the cantilever at the first-order resonance frequency (external sound pressure = 1 Pa).

**Figure 3 sensors-22-06948-f003:**
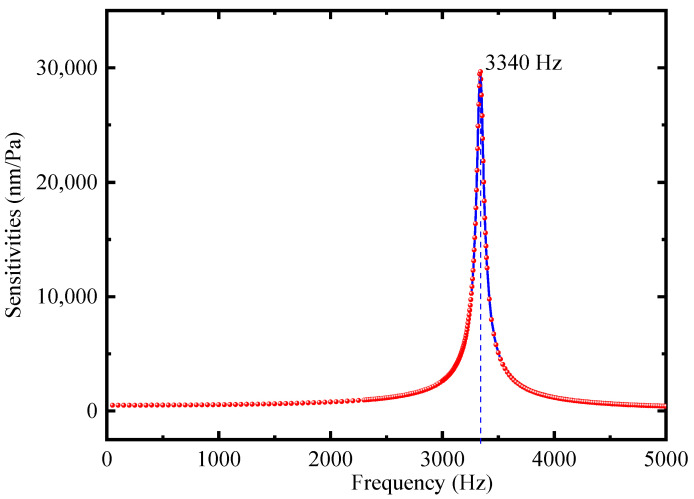
Simulation of the frequency response.

**Figure 4 sensors-22-06948-f004:**
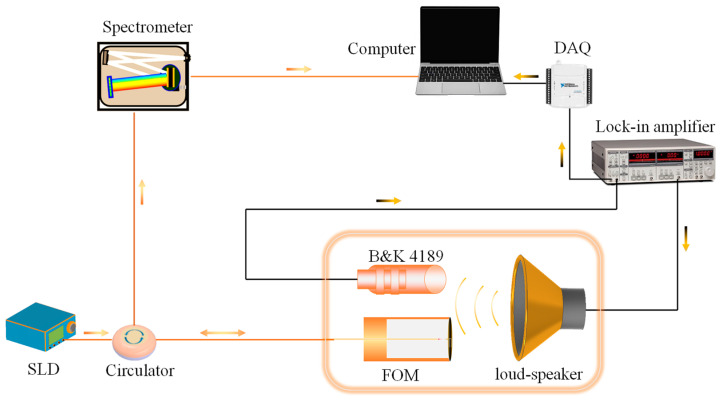
The principle diagram of the experimental setup for the designed FOM.

**Figure 5 sensors-22-06948-f005:**
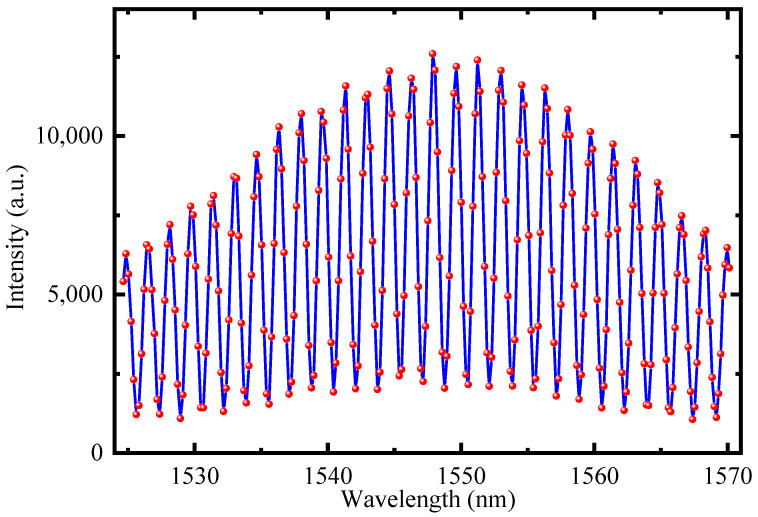
Interference spectrum of the FOM.

**Figure 6 sensors-22-06948-f006:**
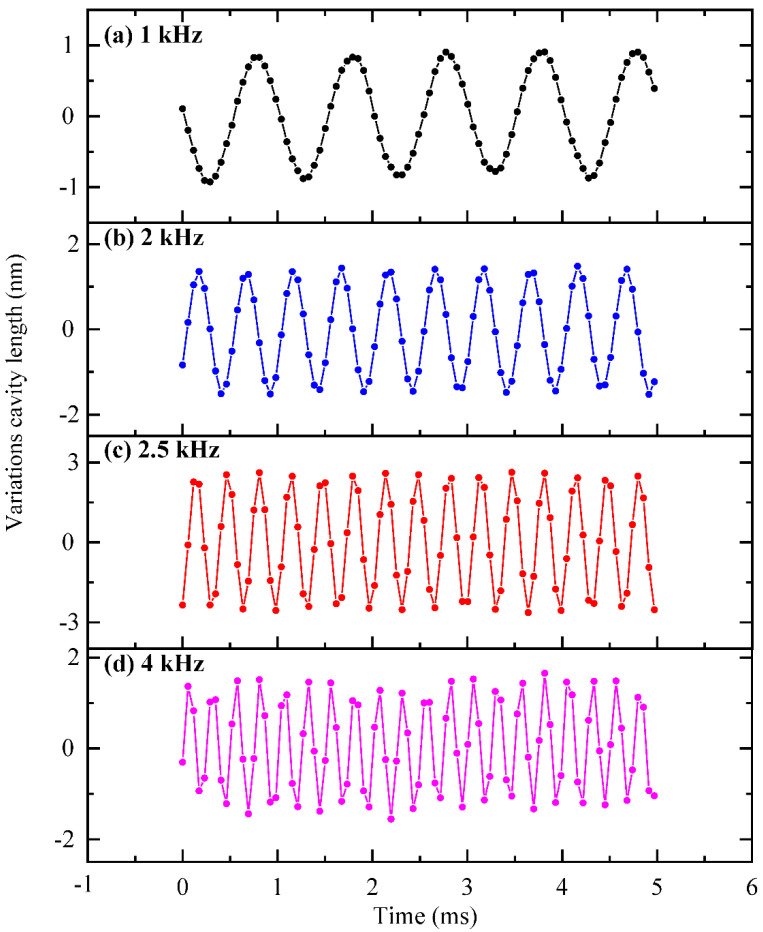
Time-domain responses of the FOM under 0.003 Pa radial acoustic pressure of (**a**) 1 kHz, (**b**) 2 kHz, (**c**) 2.5 kHz, and (**d**) 4 kHz.

**Figure 7 sensors-22-06948-f007:**
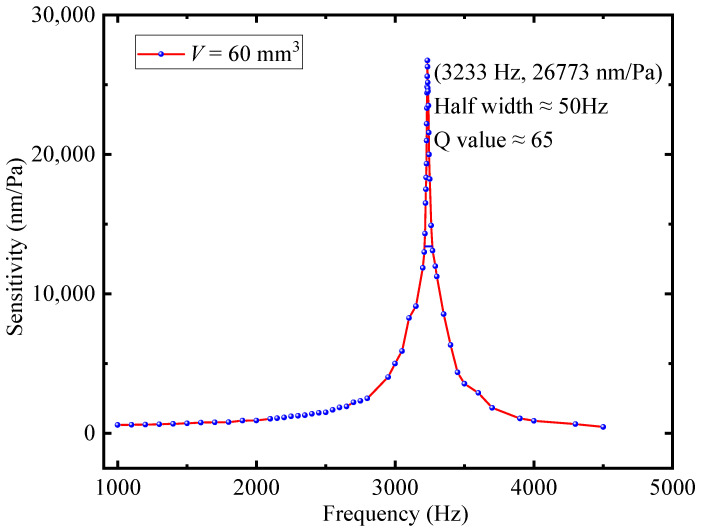
Frequency response of the designed FOM.

**Figure 8 sensors-22-06948-f008:**
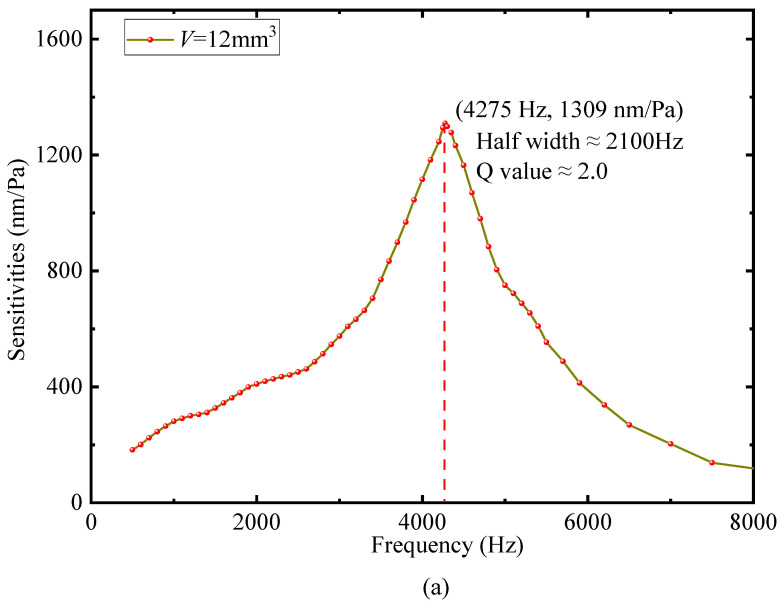
The FOM experimental frequency response curve of the different cavity volumes; (**a**) 12 mm^3^; (**b**) 30 mm^3^.

**Figure 9 sensors-22-06948-f009:**
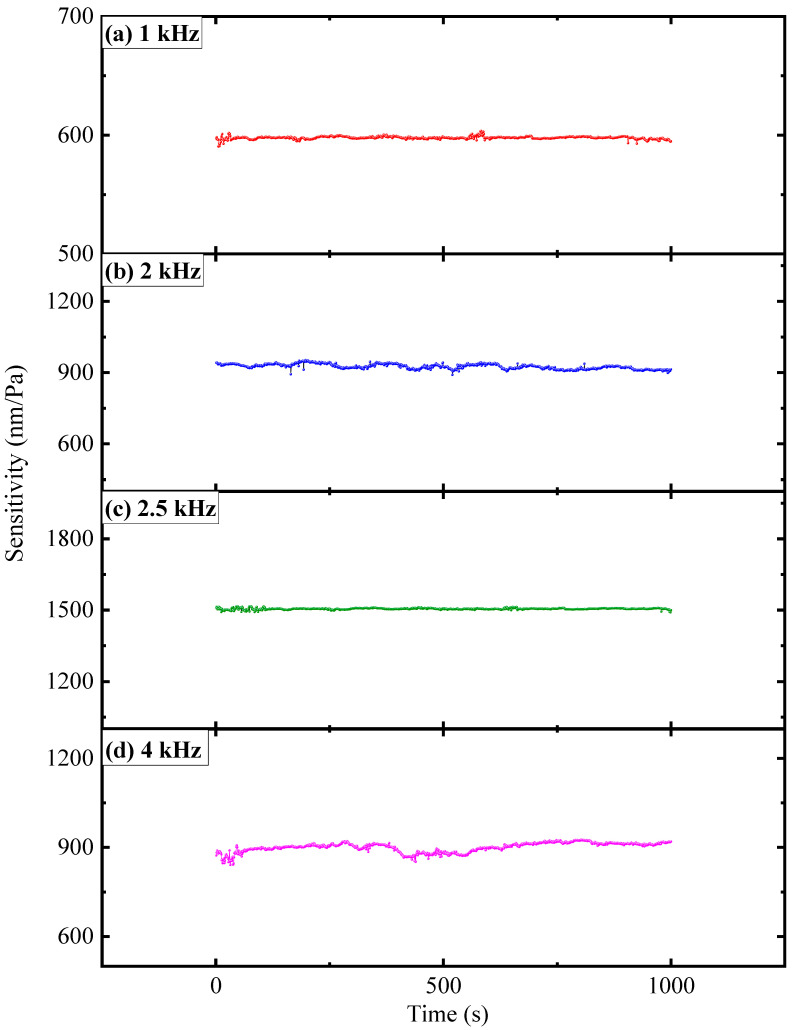
The long-term sensitivity of the FOM at different operating frequencies.

**Figure 10 sensors-22-06948-f010:**
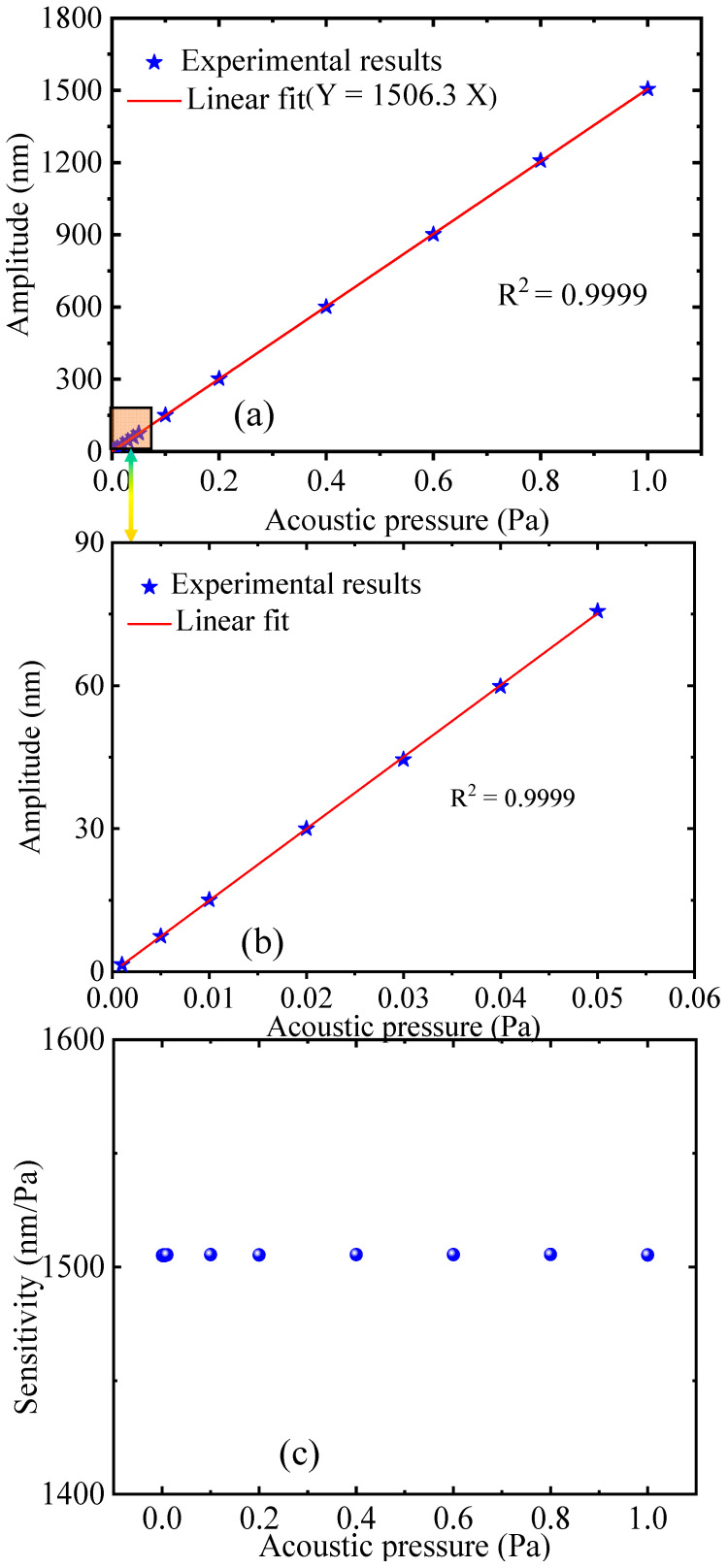
(**a**) The output amplitudes of the designed FOM under different sound pressures; (**b**) the enlarged view of the output amplitude of the designed FOM at 0.001–0.005 Pa; (**c**) the sensitivity at different sound pressures.

**Figure 11 sensors-22-06948-f011:**
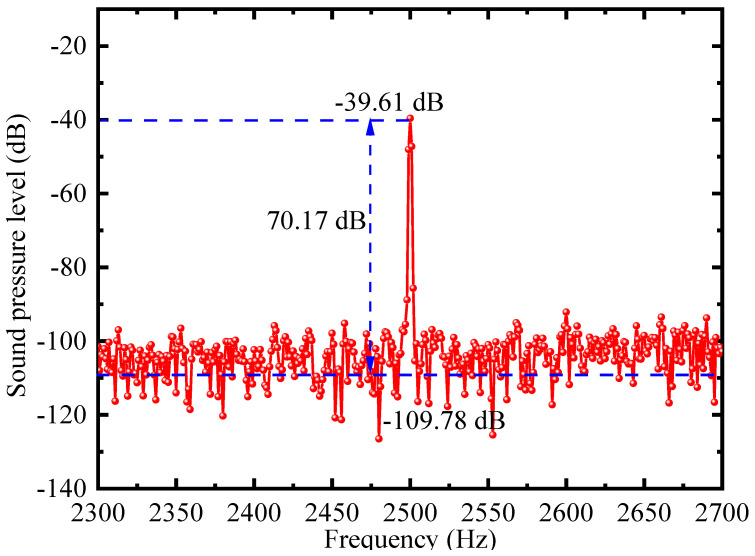
Spectrum of the FOM when a sound pressure of 0.003 Pa is applied at 2.5 kHz.

**Table 1 sensors-22-06948-t001:** Comparison of FOMs of different cavities with the same cantilever size.

Cavity Volume (mm^3^)	First-Order Resonance Frequency (Hz)	Sensitivity (nm/Pa)	Q Value
12	4275	1309	2
30	3520	7448	5
60	3233	26,773	65

**Table 2 sensors-22-06948-t002:** Long-term sensitivity analysis.

Frequency (Hz)	Average Sensitivity Value (nm/Pa)	Deviation
1000	597.7	1.4%
2000	925.7	1.1%
2500	1506.1	1.9%
4000	900.6	2.3%

**Table 3 sensors-22-06948-t003:** Performance of previously developed FOM, based on FPI.

Type	Volume (mm^3^)	Sensitivity (nm/Pa)	MDP (μPa/Hz^1/2^)	Demodulation Algorithm
Silver diaphragm-based FOM [7]	--	2100	14.5 @ 4 kHz	Phase demodulation
Graphene diaphragm-based FOM [19]	--	28.57	83 @ 3 kHz	Phase demodulation
Flywheel-like diaphragm-based FOM [18]	216	1.525	13.06 @ 4 kHz	Intensity demodulation
Parylene-C diaphragm-based FOM [24]	--	2239	22.1 @ 20 Hz	WLI
Photonic-crystal membrane-based FOM [33]	191	--	1 @ 23 kHz	Phase demodulation
Cantilever-based FOM [32]	2653	364	8.5 @ 1 kHz	WLI
Silicon cantilever-based FOM [6]	1570	28,750	0.21 @ 3 kHz	WLI
Silicon cantilever-based FOM [30]	18	950	25.68 @ 13 kHz	WLI
**Silicon cantilever-based FOM (this paper)**	**102**	**26,773**	**0.93 @ 2.5 kHz**	**WLI**

## Data Availability

Not applicable.

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
