# Peer review of "Ultra-High-Sensitivity, Miniaturized Fabry-Perot Interferometric Fiber-Optic Microphone for Weak Acoustic Signals Detection"

_sensors, 2022, doi:10.3390/s22186948_

Round 1

Reviewer 1 Report

The paper entitled "Ultra-high sensitivity, miniaturized Fabry-Perot Interferometric fiber-optic microphone for weak acoustic signals detection" discloses an ultra-high sensitivity FOM.

There are interesting results but some improvement is needed.

Comments:

1. do not indent the text after a formula.

2. P2, L89: "Where m, \omega, \omega_0 are the effective mass of the cantilever, 1st angular frequency and angular frequency." I think \omega is the frequency and \omega_0 the first angular frequency?

3. Please provide more information on the finite element simulation of figure 2.

4. Indicate the sense of the circulator in figure 4.

5. mm^3 in the figure 7 and not mm_3.

6. P11, L209: "Figure 11 depicts the amplitude spectrum of the FOM with a sound pressure of 0.01 Pa is applied at 3 kHz. It shows that the floor-noise and the SNR are -117.58 dB and -25.53 dB for a 1 Hz resolution bandwidth."

According to the graph, the signal is -25.53 dB and the SNR is 92.05 dB. Please check your text.

7. P11, L212: "Table 2 summarizes ..." It should be Table 3.

8. P11, Table 3: i do not understand why the volume is 102 mm^3. It is probably better to give the volume of the cavity (60 mm^3), as it seems to be one of the main parameters. In corollary, the cavity volumes in figures 7 were 12 and 30 mm^3, but should we understand that the FOM volume was kept at 102 mm^3? Please better comment that part.

9. I probably missed one important information, but is it sufficient to get a sharp resonance with a high Q factor to make a good FOM? What is the response if the acoustic wave to be detected is not a sine wave? The resonance (and the phase response) will lead to a strong deformation of the detected signal. I think this point should be discussed in the paper with emphasis on the possible applications.    

Reviewer 2 Report

The authors present an interesting paper where a miniaturized Fabry-Perot Interferometric (FPI) fiber-optic microphone (FOM) has been developed utilizing a silicon cantilever as an acoustic transducer. The authors claim that the minimum detectable pressure (MDP) and signal-to-noise ratio (SNR) of the designed FOM are 0.25 μPa/Hz1/2 and 92 dB at an acoustic pressure of 0.01 Pa. But some major revisions should me provided:

1. The authors obtained the resonant frequency of the sensor in the region of 3233 Hz. Further, the tests of the interferometer take place just near the resonant frequency. It is quite logical that the sensor shows the attractive results this way. At the same time, in [7], to which the authors refer, to assess the sensitivity, a test frequency is selected at a relatively flat section of the acoustic pressure sensitivity vs kHz characteristic, far from resonant peaks. In [7] they have obtained 160 nm/Pa and 14.5 μPa/Hz1/2 @ 4 kHz. If there are any significant reasons for comparing these two results, it would be highly desirable that the authors add this information to the manuscipt. The same is about the SNRs.
If my fears are correct, it is recommended to repeat the comparison for as close as possible the same test conditions. If not, please give the additional explanations in the text.
2. (Not as a requirement, but as a suggestion) If the performance of the sensor is still evaluated at the resonant frequency, then the authors are advised not to use the term "microphone", replacing it with something like "acoustic indicator".
3. It would be nice if the authors added some comments to the method comparison 'Table 3. Performance of previously developed FOM based on FPI'. In particular, this applies to installations in which the sensor operated. If there are any differences in the principles of detection described in different sources (given in the table), this should be mentioned in the "Conclusions" section.
4. The Section 3.1 ends with a picture. I propose to move this picture up, immediately after its first mention in the text.

I think this paper is rather interesting and it will be published in Sensor after these revisions provided.

Round 2

Reviewer 1 Report

My comments have been correctly addressed.

Three small typos:

1. no captital letter for where after formulas (1) and (3)

2. L102 Pa and not pa for pascal unit

3. L187: data and not datas  

Author Response

Thank the reviewer for the suggestions. We have corrected it in our revised manuscript-R2. Please see the attachment.

Reviewer 2 Report

Dear authors, thank you for paying attention to all my comments. Now the paper looks nice. Good luck with the publication!

Author Response

Thank the reviewer